# Evaluation of Elastic Filament Velocimetry (EFV) Sensor in Ventilation Systems: An Experimental Study

Athanasia Keli [1,*], Samira Rahnama [1], Göran Hultmark [1], Marcus Hultmark [2] and Alireza Afshari [1,*]

1 Department of the Built Environment, Aalborg University, 2450 København, Denmark
2 Department of Mechanical and Aerospace Engineering, Princeton University, Princeton, NJ 08544, USA
* Correspondence: akeli@build.aau.dk (A.K.); aaf@build.aau.dk (A.A.)

**Abstract:** Determination of airflow rates is an inevitable part of the energy-efficient control of ventilation systems. To achieve efficient control, the flowmeters used must be suitably accurate and create minimum disturbance to the airflow. In this study, we evaluate the quantitative performance characteristics of an innovative micro-electromechanical systems (MEMS) flowmeter, a so-called Elastic Filament Velocimetry (EFV), in ventilation ducts. Two versions of the EFV-sensor, i.e., an 11-nanoribbon and a 22-nanoribbon variety, were evaluated in laboratory studies. The results indicate that the 11-nanoribbon sensor is more suitable for air velocity measurements in ducts than the 22-nanoribbon sensor. The 11-nanoribbon sensor can measure air velocities from 0.3 m/s. The maximum variation of the sensor-output is 3% for velocities over 0.5 m/s. Calibration models have been developed for the 11-nanoribbon sensor. The error due to model calibration is lower than ±5% for velocities over 0.6 m/s. Moreover, laboratory studies were performed to investigate the airflow disturbance in a duct system due to the EFV sensor. The results were compared with the corresponding disturbance caused by two different types of self-averaging probes. At a bulk velocity of 3 m/s, the self-averaging probes introduced a greater pressure drop by at least 50% compared to the EFV-sensor.

**Keywords:** EFV-sensor; MEMS airflow sensor; air velocity measurements; flow disturbance; ventilation systems

## 1. Introduction

In mechanical ventilation systems, energy is mainly required for fan operation. The energy use of the fan is a function of the airflow rate, pressure drop in the distribution system, and fan efficiency. A study from 2001 [1] indicated that the energy demand for fan operation can account for between 15% and 50% of the total energy required for the operation of heating, ventilation, and air conditioning (HVAC) systems. The energy demand for the operation of HVAC systems can be up to 60% of the total energy required for building operations [2]. With regard to reducing energy use from fan operation, several studies [3–6] have focused on designing energy-efficient ventilation systems and implementing efficient control strategies, aiming to minimize either the pressure drop in the system, the daily airflow rate, or both. Accurate determination of airflow rates in ventilation ducts is vital to achieving efficient control as well as to uncovering inadequate and inefficient performance of the ventilation system [7,8]. Deficiencies in monitoring and responsive control of airflow rates can affect occupant comfort, health, and well-being as well as energy use. Based on standard fan laws [9], fan power is proportional to the airflow rate to the third order. For example, a 10% airflow rate reading error could result in 33% fan energy waste. However, by itself, the airflow rate determination can directly or indirectly be an obstacle to the goal of decreasing the power demand for mechanical ventilation, as flowmeters mounted or extending into the duct disturb the airflow, causing an additional pressure drop in the distribution system. Therefore, it is a requirement that flowmeters be suitably accurate and cause minimal disturbance to the airflow.

A typical ventilation duct flowmeter is the self-averaging probe with several fixed total and static pressure sensing ports, known as an averaging Pitot tube (APT). APTs are differential pressure flowmeters that determine the airflow rate using Bernoulli's principle. On the market, they can be found in three types, i.e., single axis, two axis, and flow ring. The inaccuracy level of APTs can vary from within ±5% to more than 10% of the measured values [10]. Factors affecting the accuracy level of APTs include the range of airflow rates, i.e., low airflow rates increase uncertainty [10,11], the airflow profile, which depends on the distance from airflow disturbances [10,11], the range of introduced differential pressure, which depends on the shape and cross sectional area of the probes [12–14], and the accuracy of the differential pressure transducer used for the pressure measurements [10]. Due to the small range of the introduced differential pressure, APTs are usually used to measure air velocities greater than 1 m/s. Moreover, although APTs introduce a lower permanent pressure drop in the distribution system compared to, e.g., Venturi or orifice plate flowmeters [12,14], they can nonetheless increase the total pressure loss significantly.

Ultrasonic flowmeters (transit time acoustic flowmeters) are commonly used as an alternative to APTs for airflow rate monitoring in ventilation ducts. These types of flowmeters determine the average air velocity along the path of an emitted beam of ultrasound. The average air velocity is determined by averaging the difference in measured transit time between the pulses of ultrasound propagating into and against the direction of the flow. Ultrasonic flowmeters typically have two clamp-on and wetted transducers in a W-type, V-type, or Z-type arrangement. The only difference between the three types is the propagation distance of the ultrasonic wave. One of the advantages of ultrasonic flowmeters is that they are non-intrusive and do not introduce an additional pressure drop into the distribution system. The inaccuracy level of ultrasonic flowmeters is approximately ±5% of the measured values [15], which increases at low airflow rates, i.e., velocities lower than 1 m/s. The factors that mostly affect the accuracy level of the ultrasonic flowmeters are the installation method of the transducers, i.e., the distance [16,17] and number of paths [18] between transducers, the measurement of the transit time, especially for ducts with a small inner diameter [19], and the velocity distribution across the duct, i.e., the airflow profile [20–22].

In summary, the main disadvantages of these two frequently used duct flowmeters are their relatively high inaccuracy level (±5%), which increases even more at low airflow rates, and the requirement that the airflow at the measurement plane be fully developed in order to achieve the highest possible accuracy, which is a difficult requirement to fulfil in practice. Flowmeters are commonly installed close to airflow disturbances, e.g., bends, diameter changes, etc., which affect measurement accuracy. Furthermore, these two frequently used types of flowmeters are bulky, power hungry, and quite expensive.

Recently, efforts have been made to introduce flow sensors based on MEMS technology into ventilation systems in order to reduce excessive energy use through more accurate control of airflow rates. For example, Shikida et al., 2012 [23] and Shikida et al., 2013 [24] introduced a flexible MEMS thermal airflow sensor with low power consumption (12.2 mW) for application in large-scale air conditioning network systems in buildings. The sensor was composed of four thin hot-film anemometers attached at 90° angles inside the surface of a duct. The sensor was calibrated as such that it could determine the airflow rate in straight ducts as well as downstream of bend ducts and butterfly dampers that have different angles. The sensor could measure air velocities up to 3000 m$^3$/h (corresponding to 26 m/s). However, the sensor repeatability was limited (5%) for airflow rates lower than 300 m$^3$/h (corresponding to 2.6 m/s). In addition, the sensor suffered from signal drifting issues due to ambient temperature effects. In another study, Glatzl et al., 2016 [25] presented a thermal airflow sensor for HVAC systems based on printed circuit board (PCB) technology. Their target was to develop a flexibly designed and cost-effective airflow sensor that could measure the air velocity over the whole cross-section of a ventilation duct. The sensor was based on a calorimetric operation principle. CFD simulations indicated that the sensor could be used for velocities up to 15 m/s. However, laboratory studies were conducted

only for an air velocity range within 1 m/s and 3.66 m/s. One drawback was that this sensor required more than 100 mW of power to generate a prominent sensor signal. Xu et al., 2020 [26] presented a thermoresistive microcalorimetric flow (TMCF) sensor based on complementary metal-oxide semiconductor (CMOS) MEMS technology. The sensor had a measuring range between −23 m/s up to 23 m/s and required less than 4.2 mW of power to generate a prominent sensor signal. The sensor achieved an accuracy of ±0.05 m/s (with a linear fitting solution) within a flow range of −2 m/s to 2 m/s, while the minimum detectable flow (MDF) was 2.5 mm/s. One drawback of the sensor was that the TMCF sensor required a fully developed flow region to accurately determine the airflow rate in a ventilation duct.

The aim of the present study is to extend the use of MEMS flow sensors in ventilation systems and contribute to the reduction of energy use and improvement of indoor environments through suitably accurate and low-resistance measurement of airflow rates. MEMS flowmeters are superior to traditional ones thanks to their high accuracy, small size, low cost, high sensitivity, high temporal and spatial resolution, low power consumption, and batch fabrication compatibility [27]. Therefore, the present study evaluates the quantitative performance characteristics of an innovative MEMS flow sensor, a so-called EFV, in ventilation ducts. To date, EFV sensors have been used to measure liquid velocity; however, in the present study it is modified to measure air velocity in ventilation ducts. To the best of our knowledge, this is the very first time such a sensor has been tested. Laboratory studies were conducted to evaluate the quantitative performance characteristics of the EFV sensor in ventilation ducts. Furthermore, laboratory studies were performed to investigate the airflow disturbance in the duct system due to the presence of the EFV-sensor compared to other types of flowmeters.

In the rest of this paper, we first briefly describe the EFV sensor. Next, the methodology used to evaluate the sensor's performance in the ventilation ducts and its influence on the airflow is presented along with the most relevant results. Finally, drawbacks and related future works are discussed.

## 2. Materials and Methods

### 2.1. EFV Sensor

The EFV sensor is a prototype developed at Princeton University, USA, for measuring fluid velocity. Thus far, it has been tested and commercialized mainly for liquid applications. The original version of the EFV sensor is a MEMS strain-based sensor. The EFV sensor uses free-standing electrically conductive nanoribbons suspended between silicon substrates. The platinum nanoribbons have dimensions of 0.5 mm length, 6.5 μm width, and 150 nm thickness; the thickness of human hair is about 500 times greater than the thickness of such a nanoribbon. The structure and shape of the sensor resemble the design of a nanoscale hot-wire sensor, while its operation is similar to that of a strain gauge. The change in resistance of the nanoribbons due to the strain is measurable by a simple Wheatstone bridge, and can be directly correlated to the flow velocity. These features result in a very low-cost and efficient measuring device. For a more in-depth review of the EFV sensor, see Fu et al., 2017 [28]. The high-frequency response of the sensor (it can have a bandwidth of up to 100 kHz) together with the nanoscale dimensions of its sensing elements make the EFV sensor suitable for measuring turbulence fluctuations. Laboratory studies have shown that the EFV sensor can be used to measure streamwise and wall normal turbulence fluctuations [29,30].

Figure 1 shows the layout of (a) the EFV sensor's sensing elements and (b) the EFV sensor's chip; the blue rectangle is the through-hole of the sensor, which is spanned by the nanoribbons. The dimensions of the rectangular through-hole are 0.5 mm width, 1 mm length, and 0.5 mm depth. The Wheatstone bridge used to measure the change in resistance of the nanoribbons is shown on the sensor chip (Figure 1b). The sensor chip is mounted on the edge of a like-stick printed circuit board (PCB) with dimensions of 4 mm width, 1.6 mm

thickness, and customized length. The PCB-stick is mounted on a larger PCB (Figure 2). The PCB-stick has the same size through-hole.

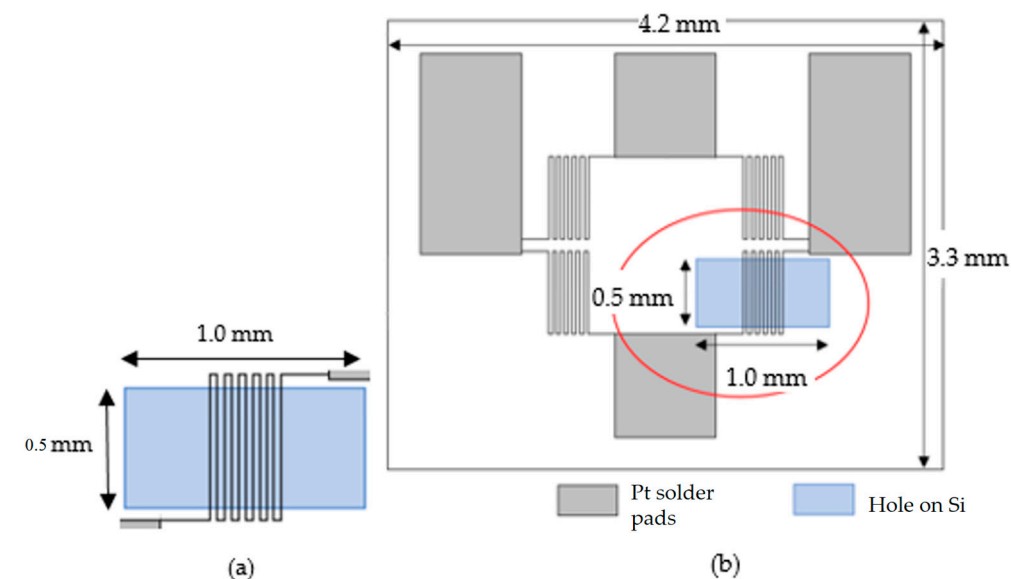

**Figure 1.** EFV sensor layout: (**a**) sensor through-hole spanned by nanoribbons and (**b**) top-down view of EFV chip.

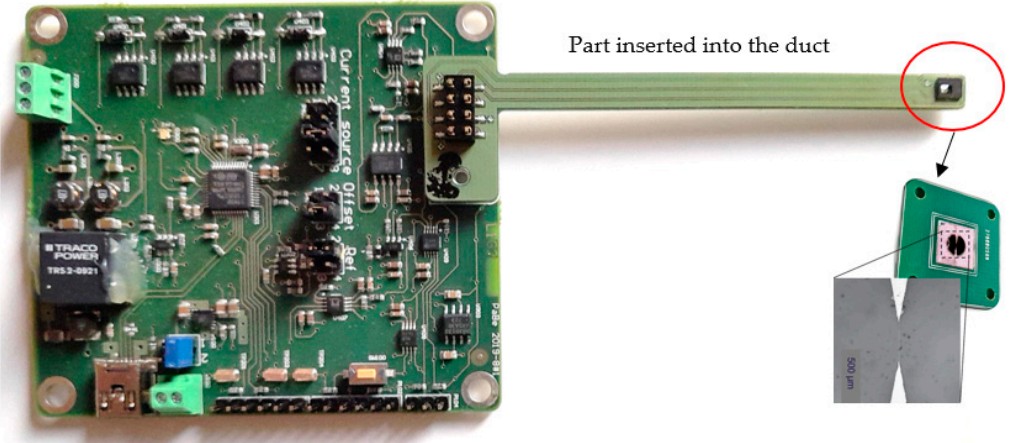

**Figure 2.** EFV chip, PCB-stick, and large PCB.

In this project, the EFV sensor has been modified to measure air velocities in ventilation ducts. To the best of our knowledge, this is the first time the EFV sensor has been used in ventilation ducts, i.e., free flow conditions. Two EFV sensor versions have been developed, a 22-nanoribbon sensor and an 11-nanoribbon sensor. The 22-nanoribbon sensor, with a resistance of 1800 ohms, requires 2 mA circuit current (8 mW), while the 11-nanoribbon sensor, with a resistance of 1000 ohms, requires 4 mA circuit current (16 mW). Both take all of their power from a standard USB connected to a PC. The entire research circuit requires less than 100 mW. A stable signal is ensured using internal filters and power management components, for which an integrated 16-bit A/D converter is used. The data are sent through a standard USB connection. The measurement range of the A/D converter is 3.3 V, while the resolution is ≈50.4 uV. The sampling frequency of the sensors is 100 Hz.

### 2.2. Laboratory Studies

#### 2.2.1. EFV Sensor Characterization

Laboratory studies were conducted to evaluate the response of the EFV sensors, i.e., the 11-nanoribbon and 22-nanoribbon varieties, in relation to different air velocities and temperatures. The influence of the air velocity on sensor output was evaluated by conducting repetitive tests, during which the air velocity varied from 0 m/s up to 4 m/s (upscale testing) while the temperature was kept constant at 23 °C (reference temperature) with a maximum variation of 0.05 °C. Measurements at each reference point lasted 1 min. In another set of tests, the influence of temperature variations on the output of the EFV sensors was investigated by varying the temperature from approximately 23 °C up to 29 °C (upscale testing) while the flow rate was kept constant. The EFV sensor output was constantly monitored as the temperature increased. Note that auto-calibration of the sensors was carried out at zero pressure on the sensing elements each time the electronics were turned on in order to correct the zero-offset. A warm-up period of 5 min was assumed before beginning to log the data.

The experimental setup used to characterize the EFV-sensors is shown in Figure 3. The test rig consisted of an axisymmetric tunnel (80 mm diameter) used to create a free jet with uniform velocity profile and a low level of turbulence. The EFV sensor was inserted into the jet centerline. As the jet centerline velocity was known, the sensor output could be evaluated in relation to a reference velocity. The airflow was regulated by changing the voltage input to the fan, which consequently changed the fan speed. To determine the air velocity, two different orifice plates with diameters of 23 mm and 46 mm were used as reference instruments (Bernoulli's principle). The pressure difference across the plates was monitored using a type FCO510 micromanometer from Furness. Then, the air velocity was calculated using the corresponding calibration curves of the orifice plates. During the tests in the laboratory environment, the barometric pressure and temperature were monitored in order to correct the calculated velocity for these parameters, as the calibration curves of the orifice plates were developed at an air density of 1.2 kg/m$^3$. A type 2104 barometer from Mensor was used to measure the barometric pressure. The temperature was measured using recently-calibrated thin thermocouples, which were inserted into the jet centerline 2 cm downstream of the EFV sensor as well. Details about the measuring devices of the calibrator can be found in Table 1. The reference velocity was calculated using Equations (1) to (4). The expanded uncertainty of the reference velocity was 0.3%, calculated based on GUM [31].

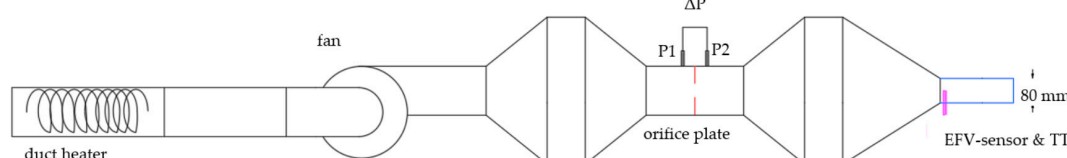

**Figure 3.** Test rig used for EFV sensor characterization.

The calibration curves of the orifice plates with 23 mm and 46 mm diameter, respectively, are provided in Equations (1) and (2). The corrected air density based on the measured temperature is provided in Equation (3). The corrected air velocity is provided in Equation (4).

$$v = 0.744 \, \Delta P^{0.4516}, \tag{1}$$

$$v = 2.886 \, \Delta P^{0.49}, \tag{2}$$

$$\rho_{air,corr} = P_{barometric} / (287.04 (T + 273)), \tag{3}$$

$$v_{corr} = v \, (\rho_{air} / \rho_{air,corr})^{1/2}, \tag{4}$$

where v is the air velocity in m/s, $\Delta P$ is the pressure difference in Pa, $P_{barometric}$ is the barometric pressure in Pa, T is the air temperature in °C, and $\rho_{air}$ is the air density in kg/m$^3$.

**Table 1.** Characteristics of measuring devices of calibrator.

| FCO510 Micromanometer [32] | |
|---|---|
| Measuring range: | 0 Pa to 200 Pa |
| Accuracy: | ±0.25% of reading value between 20 Pa and 200 Pa |
| Resolution: | 0.001 Pa |
| Mensor Digital pressure gauge [33] | |
| Calibrated range: | 10.8 psia to 16.7 psia |
| Temperature compensated: | 15 °C to 45 °C |
| Accuracy: | ±0.01% of reading value |
| Resolution: | 1 Pa |
| Thin thermocouple (TT) | |
| Accuracy: | ±0.05 °C |
| Resolution: | 0.01 °C |

### 2.2.2. Airflow Disturbance Studies

Laboratory studies were performed in order to investigate the airflow disturbance in a duct system due to the presence of different flowmeters, i.e., the EFV sensor and two different types of APTs, so-called measuring crosses of Type I and Type II. The airflow disturbance was investigated by measuring the pressure drop by conducting repetitive tests, during which the airflow rate was varied from 60 L/s up to 141 L/s corresponding to a bulk velocity of 3 m/s up to 7 m/s (upscale testing), while the temperature was kept constant with a maximum variation of 0.05 °C. Each measurement lasted 5 min with a time interval of 1 s.

The test rig used for the pressure drop studies is presented in Figure 4. The airflow rate was regulated using an UltraLink FTCU from Lindab, which is an ultrasonic flowmeter attached to a damper body. The EFV sensor and the measuring crosses were used only as obstacles to investigate their disturbance to the airflow. Figure 5 presents the measurement plane, Figure 5a presents a duct with the EFV sensor mounted inside, and Figure 5b,c presents ducts with measuring crosses of Type I and Type II, respectively. The EFV sensor extended to the duct's center. The flowmeter under investigation was installed in a fully developed flow region. The pressure drop was determined by measuring the static pressure. Thus, pressure taps were soldered to the ducts. The measurement points remained identical for all study cases. A type 480 pressure differential transducer from Testo was used for the pressure measurements. The temperature was measured using UltraLink FTCU. Details about the measuring devices used in the pressure drop studies can be found in Table 2.

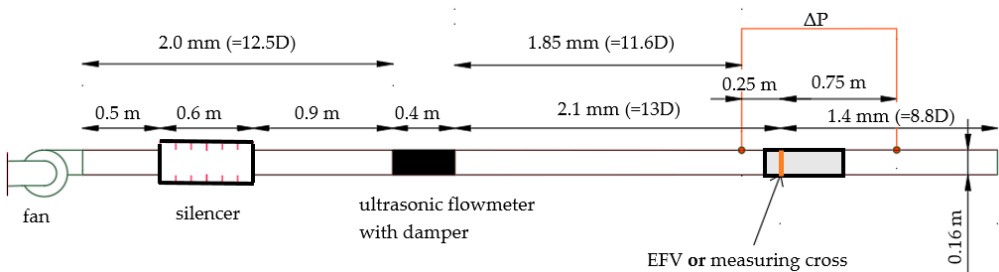

**Figure 4.** Test rig used for pressure drop studies.

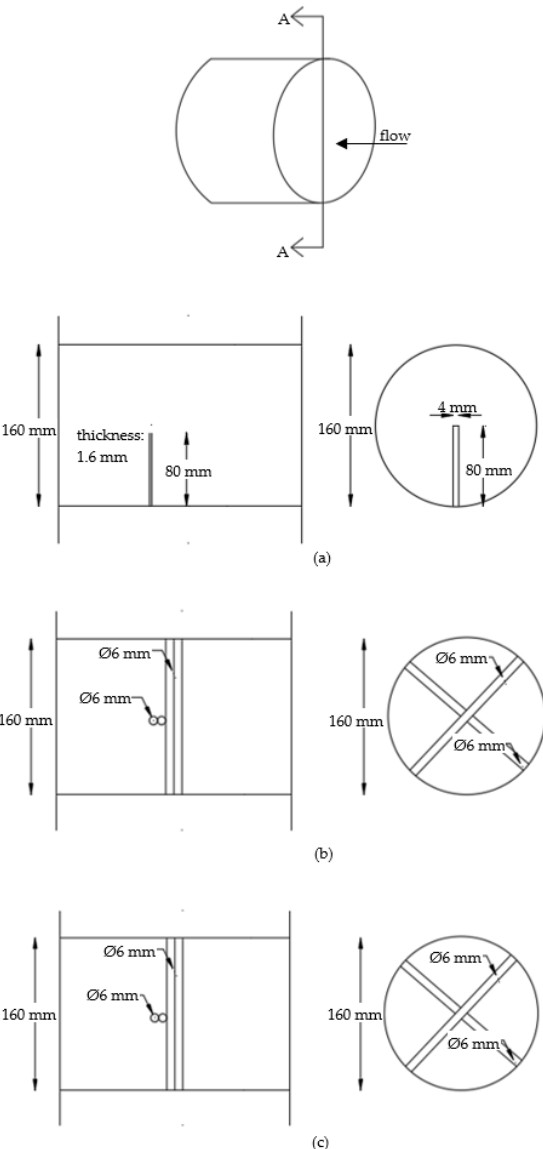

**Figure 5.** Layout of measurement plane: (**a**) cross-section A-A of duct with EFV sensor (**Left**) and front view of duct with EFV sensor (**Right**); (**b**) cross-section A-A of duct with measuring cross Type I (**Left**) and front view of duct with measuring cross Type I (**Right**); (**c**) cross-section A-A of duct with measuring cross Type II (**Left**) and front view of duct with measuring cross Type II (**Right**).

**Table 2.** Characteristics of measuring devices used for pressure drop investigation.

| UltraLink FTCU [34] | |
|---|---|
| Airflow rate measuring range: | 4 L/s to 302 L/s |
| Airflow rate accuracy: | ±5% of reading value OR±1.6 L/sbased on which of these two values is the greatest |
| Temperature measuring range: | −10 °C to 50 °C |
| Temperature accuracy: | ±1 °C |
| Testo 480 [35] | |
| Measuring range: | −100 hPa to 100 hPa |
| Accuracy: | ±(0.3 Pa + 1% of reading value) |
| Resolution: | 0.1 Pa |

## 3. Results and Discussion

### 3.1. EFV Sensor Response to Air Velocity

To evaluate the response of the different versions of the EFV sensor to air velocity, the results from the repetitive tests conducted under the same conditions are presented in Figures 6 and 7. The 11-nanoribbon sensor is able to measure lower velocities than the 22-nanoribbon sensor (Figure 6), as it has almost half the resistance (1000 ohms instead of 1800 ohms). Specifically, the 11-nanoribbon sensor can measure air velocities from 0.3 m/s, while the 22-nanoribbon sensor can measure air velocities from 0.5 m/s. Moreover, it can be seen in Figure 7 that thanks to its lower resistance, the 11-nanoribbon sensor has higher sensitivity compared to the 22-nanoribbon sensor, e.g., at 1.5 m/s velocity, the output sensitivity of the 22-nanoribbon sensor is ≈103 mV/(m/s), while the corresponding sensitivity of the 11-nanoribbon sensor is ≈350 mV/(m/s). Furthermore, both sensor versions have high precision. For velocities over 0.5 m/s, the maximum variation of the signal of the 22-nanoribbon sensor is 5%, while the corresponding variation of the 11-nanoribbon sensor is 3%.

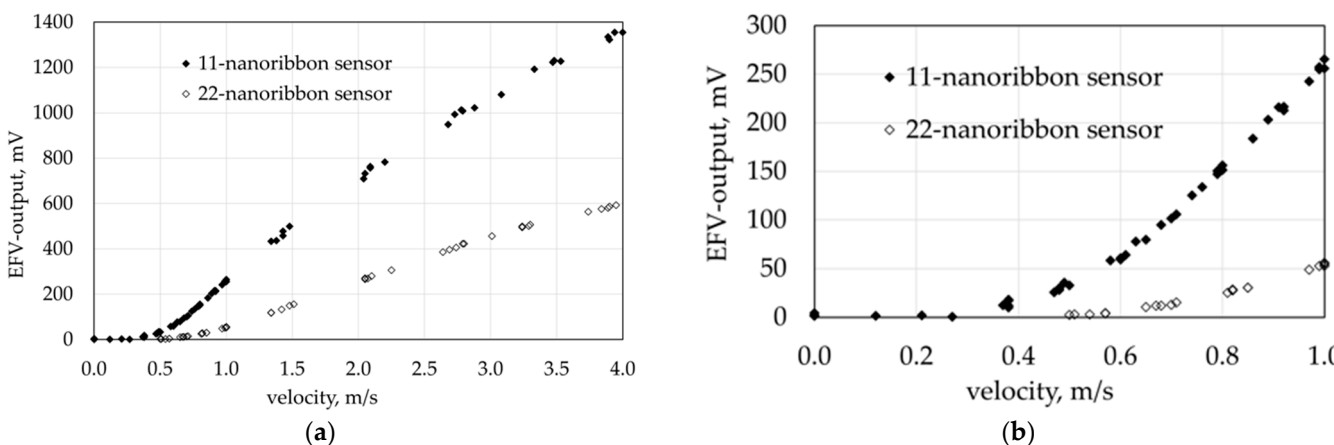

**Figure 6.** EFV sensor output as a function of air velocity—upscale testing at reference temperature of 23 °C: (**a**) full range and (**b**) zoomed-in view of low velocities.

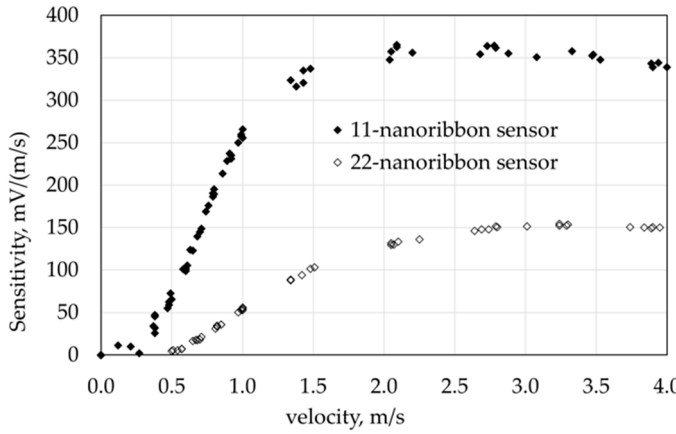

**Figure 7.** Sensitivity of EFV sensor as a function of air velocity—upscale testing at a reference temperature of 23 °C.

### 3.2. EFV Sensor Response to Temperature

The results from the tests conducted to evaluate the influence of temperature variations on the output of the EFV sensors, i.e., the 11-nanoribbon and 22-nanoribbon varieties, are presented in Figure 8. Due to practical limitations, the temperature influence on the EFV sensors' output was evaluated for velocities over 1.0 m/s. Note that the sensitivity of

the EFV sensors to temperature was a known problem [28], and measures were taken at the design stage in order to compensate for the sensitivity of the sensor output to the temperature. Nevertheless, the laboratory results show that the sensor output remained influenced by the temperature. The 22-nanoribbon sensor, due to its higher resistance, is slightly less sensitive to temperature compared to the 11-nanoribbon sensor.

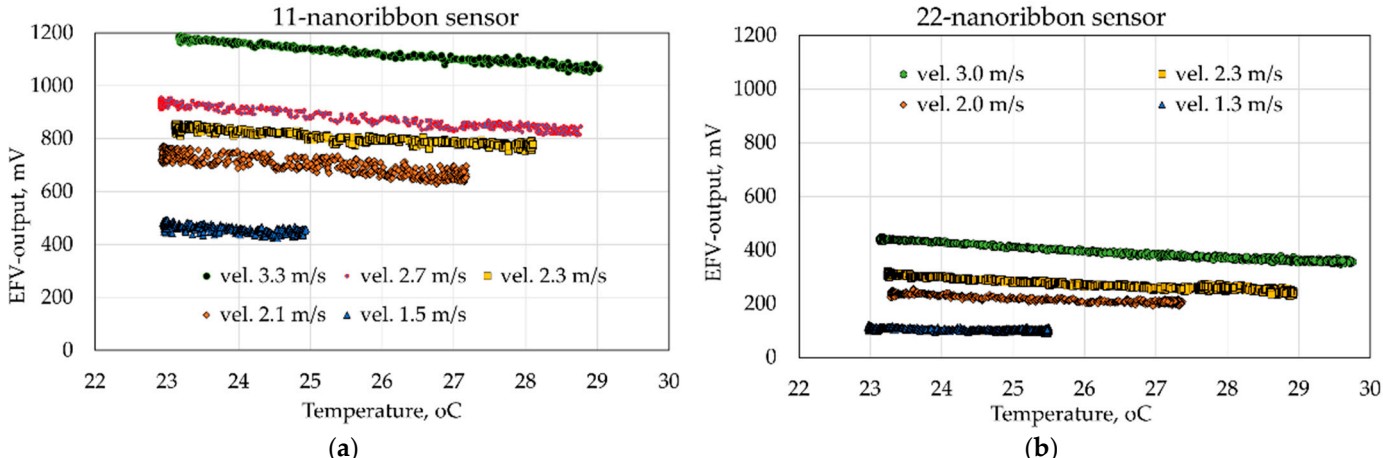

**Figure 8.** EFV sensor output as a function of temperature at different constant air velocities: (**a**) 11-nanoribbon sensor and (**b**) 22-nanoribbon sensor.

### 3.3. Development of Calibration Models

Calibration models were developed only for the 11-nanoribbon sensor. Based on its precision, measuringrange, and sensitivity to air velocity, we concluded that the 11-nanoribbon sensor better satisfies the requirements for airflow rate determination in ventilation ducts compared to the 22-nanoribbon sensor.

In order to take the temperature effects into consideration, it is necessary to correct the output of the 11-nanoribbon sensor using the linear model described in Equation (5). RMSE of residuals is ±12 mV.

$$\text{EFV-output\_corr} = \text{EFV-output\_meas} + 17.55 \; \Delta T, \tag{5}$$

where EFV-output_corr is the corrected signal of the sensor in mV, EFV-output_meas is the measured signal in mV, $\Delta T$ is the difference between the measured temperature and the reference temperature (23 °C) in °C, and 17.55 is a coefficient.

Following the functional relationship between the output of the 11-nanoribbon sensor and the air velocity at the reference temperature of 23 °C, this can be described by two mathematical models. For very low sensor outputs, i.e., up to 40 mV, a linear model can be used (Figure 9a), while for sensor outputs over 40 mV and up to 1400 mV a third-order polynomial model is more appropriate (Figure 9b). However, note that the presence of outliers can considerably affect the results of nonlinear analysis, and the polynomial model cannot be used with relative accuracy outside of the considered sensor output range. The RMSE values for the linear model and the third-order polynomial model are ±0.02 m/s and ±0.03 m/s, respectively. When using the EFV sensor to measure air velocity, the measurement error due to the calibration models developed at the reference temperature is a function of the air velocity (Figure 10). For velocities over 0.6 m/s, the measurement error is lower than ±5%.

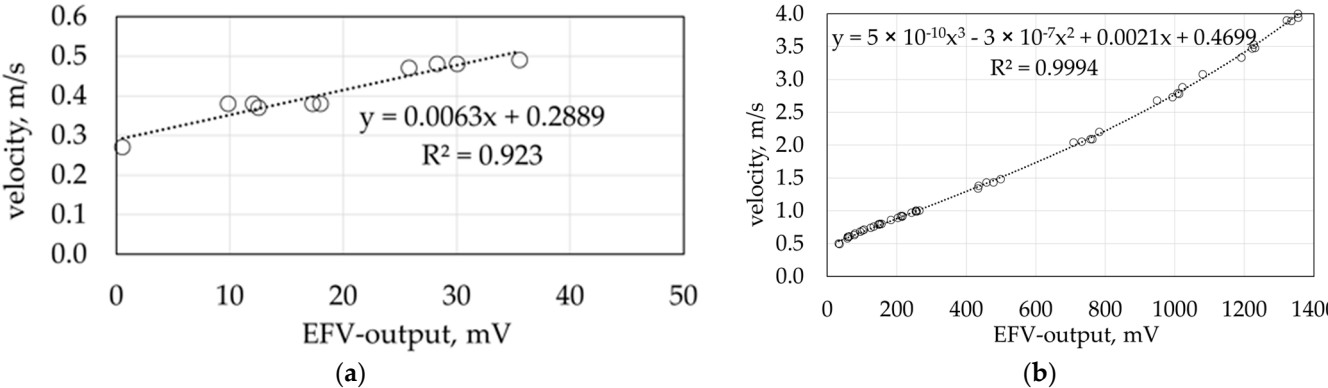

**Figure 9.** Functional relationship between the output of the 11-nanoribbon sensor and the air velocity at a reference temperature of 23 °C: (**a**) sensor outputs up to 40 mV and (**b**) sensor outputs over 40 mV and up to 1400 mV.

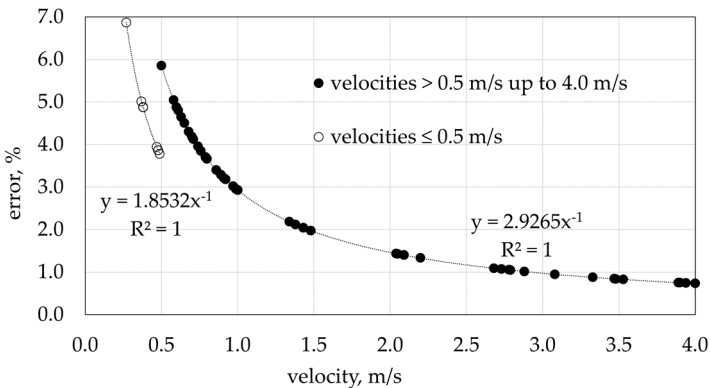

**Figure 10.** Measurement error of the 11-nanoribbon sensor due to model calibration at a reference temperature of 23 °C.

Note that for the original version of the EFV sensor (strain-based sensor), the functional relationship between the output of the sensor and the air velocity is described by a power model [28]. However, in the case of the modified EFV sensor a power model cannot be applied. The modified EFV sensor operates with a relative high circuit current, i.e., 4 mA in the case of the 11-nanoribbon sensor, which increases the temperature of the sensing elements, and as a result affects its resistance due to Joule heating. Therefore, the output of the modified sensor depends on both the strain and cooling effect.

### 3.4. Airflow Disturbance in Duct Due to the EFV Sensor

The results from the tests conducted to evaluate the airflow disturbance in a duct system due to the presence of the EFV sensor and Type I and II measuring crosses are presented in Figure 11. For a better comparison, Table 3 presents the mean values of airflow rate, temperature, and pressure drop in each case along with the expanded uncertainties, which were calculated based on GUM [31]. The EFV sensor introduces a lower pressure drop into the duct system compared to both measuring crosses. At the lowest flow rate, corresponding to a bulk velocity of 3 m/s, the EFV sensor introduces a lower pressure drop into the system by approximately 50% and 60%, respectively, compared to the Type I and Type II measuring crosses. At higher flow rates, the measuring crosses can introduce a pressure drop as much as 80% higher than the EFV sensor. Therefore, it can be expected that by replacing the more frequently used measuring crosses with EFV sensors, the energy used for fan operation could be considerably reduced due both to more accurate airflow rate determination and lower airflow disturbance. In the present study, we investigated the worst-case scenario, in which the EFV sensor extends into the center of the duct. At a later

stage, the sensor could be calibrated such that it can be placed at a distance from the duct wall, to a maximum 20% of the duct radius, in order to achieve airflow measurements with minimal airflow disturbance.

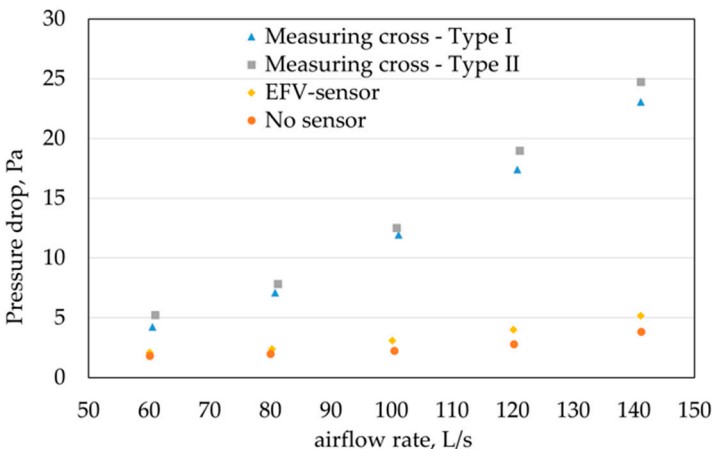

**Figure 11.** Pressure drop as a function of airflow rate.

**Table 3.** Mean values of airflow rate, temperature, and pressure drop, along with their expanded uncertainties.

|  | Airflow Rate, L/s | Temperature, °C | Pressure Drop, Pa |
|---|---|---|---|
| No sensor | 60.1 ± 4.5 | 26.6 ± 1.5 | 1.8 ± 0.5 |
|  | 80.0 ± 5.9 | 26.5 ± 1.5 | 2.0 ± 0.5 |
|  | 100.4 ± 7.5 | 26.1 ± 1.5 | 2.3 ± 0.5 |
|  | 120.1 ± 8.9 | 25.9 ± 1.5 | 2.8 ± 0.5 |
|  | 141.2 ± 10.5 | 25.9 ± 1.5 | 3.8 ± 0.5 |
| EFV-sensor | 60.0 ± 4.5 | 26.7 ± 1.5 | 2.1 ± 0.5 |
|  | 80.2 ± 6.0 | 26.4 ± 1.5 | 2.4 ± 0.5 |
|  | 100.1 ± 7.4 | 26.1 ± 1.5 | 3.1 ± 0.5 |
|  | 120.1 ± 8.9 | 25.9 ± 1.5 | 4.0 ± 0.5 |
|  | 141.1 ± 10.5 | 25.9 ± 1.5 | 5.2 ± 0.6 |
| Measuring cross, Type I | 60.5 ± 4.5 | 24.8 ± 1.5 | 4.2 ± 0.6 |
|  | 80.8 ± 6.0 | 24.3 ± 1.5 | 7.1 ± 0.6 |
|  | 101.2 ± 7.5 | 24.1 ± 1.5 | 11.9 ± 0.7 |
|  | 120.8 ± 9.0 | 24.0 ± 1.5 | 17.4 ± 0.7 |
|  | 141.1 ± 10.5 | 23.9 ± 1.5 | 23.0 ± 0.8 |
| Measuring cross, Type II | 61.0 ± 4.5 | 23.7 ± 1.5 | 5.2 ± 0.5 |
|  | 81.2 ± 6.0 | 23.4 ± 1.5 | 7.8 ± 0.6 |
|  | 100.8 ± 7.5 | 23.3 ± 1.5 | 12.5 ± 0.7 |
|  | 121.1 ± 9.0 | 23.2 ± 1.5 | 19.0 ± 0.8 |
|  | 141.1 ± 10.5 | 23.2 ± 1.5 | 24.7 ± 1.0 |

*3.5. Comparison between EFV Sensor and Ultrasonic Flowmeters and APTs*

The results from our experiments indicate that the EFV sensor is a promising alternative compared to the more frequently used duct flowmeters, as it is highly sensitive and accurate even at low flow rates, i.e., a measurement error of 5% for velocities over 0.6 m/s. Ultrasonic flowmeters and APTs have an inaccuracy level more than 10% for velocities lower than 1 m/s. Moreover, the EFV sensor disturbs the airflow less compared to measuring crosses (APTs), i.e., at 3 m/s bulk velocity, the pressure drop due to the EFV sensor is at least 50% lower than that of measuring crosses. Furthermore, the EFV sensor requires low power for operation, less than 100 mW, while ultrasonic flowmeters and APTs currently found on the market require more than 2.5 W for operation. In addition, the cost

of the EFV sensor is approximately EUR 10, which is considerably lower than the cost of the most frequently used flowmeters, which can be more than EUR 400. Note that efficient control of ventilation systems requires a sufficient number of flowmeters and actuators. Therefore, cost is especially important in the case of large-scale buildings, where multiple sensors must be installed.

## 4. Conclusions

In the present work, laboratory studies were conducted to evaluate the quantitative performance characteristics of a new MEMS flowmeter, the so-called EFV sensor, in ventilation ducts, as well as to investigate the airflow disturbance in a duct system due to the presence of different flowmeters, i.e., an EFV sensor and two different types of APTs (so-called measuring crosses). Two versions of the EFV sensor were evaluated, an 11-nanoribbon and a 22-nanoribbon type. Overall, our laboratory results indicate that the EFV sensor can be used in place of commonly used duct flowmeters, i.e., APTs and ultrasonic flowmeters, to measure air velocities in ventilation ducts without causing a significant pressure drop in the distribution system. Based on its measuring range, precision, and sensitivity to air velocity, we conclude that the 11-nanoribbon EFV sensor is more suitable for velocity measurements in ventilation ducts than the 22-nanoribbon sensor. The 11-nanoribbon EFV sensor can be used to measure air velocities from 0.3 m/s up to at least 4 m/s. The maximum variation of the signal is 3% for velocities over 0.5 m/s. The functional relationship between the 11-nanoribbon sensor's output and the air velocity at the reference temperature of 23 °C can be described by two mathematical models, i.e., a linear model with a RMSE of $\pm 0.02$ m/s for sensor outputs up to 40 mV, and a third-order polynomial model with a RMSE of $\pm 0.03$ m/s for sensor outputs over 40 mV and up to 1400 mV. As the output of the EFV sensor is sensitive to temperature changes, it must be corrected to consider the temperature in the environment by using a linear model with a RMSE of $\pm 12$ mV. The resulting measurement error due to the calibration models is lower than $\pm 5$% for velocities over 0.6 m/s. In addition, we found that the EFV sensor produces a lower pressure drop compared to measuring crosses. The pressure drop at a flow rate of 60 L/s, which corresponds to a bulk velocity of 3 m/s, was at least 50% lower than for the measuring crosses, indicating that a significant reduction in the energy used for fan operation could be achieved by replacing measuring crosses with EFV sensors.

**Author Contributions:** Conceptualization, A.K., S.R., G.H. and A.A.; Data curation, A.K.; Formal analysis, A.K.; Funding acquisition, A.A.; Investigation, A.K.; Methodology, A.K., S.R., G.H. and A.A.; Project administration, A.K.; Resources, S.R. and A.A.; Software, A.K.; Supervision, A.K., S.R., G.H. and A.A.; Validation, A.K.; Visualization, A.K.; Writing—original draft, A.K.; Writing—review and editing, A.K., S.R., G.H., M.H. and A.A. All authors have read and agreed to the published version of the manuscript.

**Funding:** The authors acknowledge the support provided by ELFORSK, a research and development program administrated by Danish Energy Association, and the Department of the Built Environment at Aalborg University. Case number: Elforsk 352-008.

**Institutional Review Board Statement:** Not applicable.

**Informed Consent Statement:** Not applicable.

**Data Availability Statement:** Data are available upon request.

**Conflicts of Interest:** The authors declare no conflict of interest. The funders had no role in the design of the study; in the collection, analyses, or interpretation of data; in the writing of the manuscript; or in the decision to publish the results.

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
