# Peer review of "Evaluation of Elastic Filament Velocimetry (EFV) Sensor in Ventilation Systems: An Experimental Study"

_sustainability, doi:10.3390/su15031955_

Round 1
Reviewer 1 Report
The paper proposed a new sensor for ventilation airflow measurements. A few questions to be addressed before publishing the paper.
1. Why it is necessary to develop this new sensor for airflow measurements?
2. What is the cost of this new sensor?
3. A comparison of the proposed new sensor with other mature products in the market should be provided.
Author Response
Thank you for taking the time to review my paper. Below you can find my point-by-point responses to your comments.
Point 1: Why it is necessary to develop this new sensor for airflow measurements?
Response 1: Determination of airflow rates is an inevitable part of the energy-efficient control of ventilation systems. To achieve efficient control the flowmeters used must be suitably accurate and create minimum disturbance to the airflow. The frequently used duct flowmeters have a high uncertainty level for velocities lower than 1 m/s (more than 10%). Furthermore, they are bulky, have a high cost, and require a lot power for operation. The EFV-sensor comes to "solve these problems". In the introduction, it is explained in more detail the need for this sensor.
Point 2: What is the cost of this new sensor?
Response 2: It is expected that the EFV-sensor will cost 10 Euros.
Point 3: A comparison of the proposed new sensor with other mature products in the market should be provided.
Response 3: The results from the experiments conducted show that the EFV-sensor is a promising alternative compared to the frequently used duct flowmeters as it is highly sensitive and accurate even at low flow rates, i.e. a measurement error of 5% for velocities over 0.6 m/s. Ultrasonic flowmeters and APTs have an uncertainty level more than 10% for velocities lower than 1 m/s. Moreover, the EFV-sensor disturbs less the airflow compared to measuring crosses (APTs), i.e. at 3 m/s bulk velocity the pressure drop due to the EFV-sensor is at least 50% lower than the measuring crosses. Furthermore, the EFV-sensor requires low power for operation, less than 100 mW while ultrasonic flowmeters and APTs found in the market require more than 2.5 W for operation. In addition, the cost of the EFV-sensor will be approximately 10 euros, considerably lower than the cost of the frequently used flowmeters, which is more than 400 euros. Note that in order to achieve efficient control of ventilation systems, a sufficiently number of flowmeters and actuators are required. Therefore, the cost is especially important in case of large-scale buildings where several sensors must be installed.
Reviewer 2 Report
Very interesting work. Looking forward to see practical applications.

Author Response
Thank you very much for taking the time to review my paper. I really appreciate your positive comments.
Round 2
Reviewer 1 Report
Thank you, the comments from last round are solved